# Research Progress of Bonding Agents and Their Performance Evaluation Methods

**DOI:** 10.3390/molecules27020340

**Published:** 2022-01-06

**Authors:** Junyan Gan, Xue Zhang, Wei Zhang, Rui Hang, Wuxi Xie, Yunfei Liu, Wen Luo, Yu Chen

**Affiliations:** 1School of Materials Science and Engineering, Beijing Institute of Technology, Beijing 100081, China; bjfu140554119@163.com (J.G.); ranm2017@163.com (X.Z.); 3220211301@bit.edu.cn (R.H.); ssclyf@bit.edu.cn (Y.L.); 2Xi’an Modern Chemistry Research Institute, Xi’an 710065, China; xiewuxi@163.com; 3Research Institute, Liaoning Qingyang Special Chemical Engineering Co., Ltd., Liaoyang 111002, China; luowenjj@163.com

**Keywords:** propellant, bonding agents, bonding mechanism, evaluation method

## Abstract

Bonding agents are an important type of additive that are used to increase the interfacial interaction in propellants. A suitable bonding agent can prevent the dewetting between the oxidant and binder, and thus effectively improve the mechanical properties of the propellant. In the current paper, the bonding mechanisms and research progress of different types of bonding agents such as alcohol amine bonding agents, borate ester bonding agents, aziridine bonding agents, hydantoin bonding agents, neutral polymer bonding agents, and so on, are reviewed and discussed. The evaluation methods of their bonding performances including molecular dynamic simulation, contact angle method, in situ loading SEM, characterization analysis, and mechanical analysis are summarized to provide design ideas and reference for future studies.

## 1. Introduction

Composite solid propellant is an energetic composite material composed of a polymer matrix (binder), additives (plasticizers, curing agents, bonding agents), and solid fillers (energetic oxidants, metal particles). To achieve high-energy composite materials, a variety of strategies have been developed and increasing the solid content is a very effective technical means [1]. However, since the propellant is a heterogeneous composite elastomer, the solid filler tends to separate from the binder matrix as loaded, resulting in serious “dewetting” at the two-phase interface. This dewetting can have significantly negative impacts on the mechanical properties, aging properties, and safety performance of the composite solid propellant.

The bonding agent is usually used to enhance the affinity between the components of the solid propellant as one of the primary components. One end of the bonding agent molecule interacts with the solid filler surface to form a layer of high tearing modulus, and the other end binds the binder system through a chemical reaction to strengthen the bonding of the interface layer and improve the mechanical properties of the propellant [2,3]. Different kinds of bonding agents have been developed for propellant systems composed of different oxidants and binders. Alkanolamine bonding agents, polyamine bonding agents, aziridine bonding agents, borate ester bonding agents, hydantoin bonding agents, and other traditional small molecular bonding agents as well as neutral polymer bonding agents (NPBA), amide tree bonding agents, hyperbranched polyether bonding agents, and other polymer based bonding agents are the commonly used bonding agents. The innovation of bonding agents has closely followed the development trend of propellants. Selecting the appropriate bonding agent and modifying its structure by introducing functional groups can improve the application performance of the bonding agent in the propellant and the performance level of the propellant.

In the current paper, the classification of commonly used bonding agents and the evaluation methods of their bonding performance are sorted out to clarify the blending effect of bonding agent on the properties of composite solid propellant. By reviewing the mechanisms and application scopes of different bonding agents and various evaluation methods of the interfacial bonding between bonding agent and matrix filler, we summarize the strategies improving the qualitative and quantitative analyses of bonding agents.

## 2. Progress of Different Types of Bonding Agents

### 2.1. Alcohol Amine Bonding Agent

Alcohol amine bonding agents are a class of small molecule compounds of amine containing hydroxyl groups [4]. The general structure formula of an alcohol amine bonding agent is shown in Table 1. In addition to bonding functions, diethanolamine and triethanolamine also show good dispersing and wetting properties. Alcohol amine bonding agents were initially used in polyurethane (PU) propellants. Chen [5] synthesized a hydroxyl terminated alcohol amine polyester bonding agent using N-methyldiethanolamine and sebacic acid as the raw materials. The bonding agent was able to improve tensile strength and elongation of the AP/RDX/HTPB propellant by 20% and 40%, respectively. Figure 1 shows the bonding mechanism of the alcohol amine bonding agent. It reacts with amine perchlorate (AP) to form an ammonium salt ion bond, and thus is strongly bound with the oxidant surface. The hydroxyl group reacts with the binder and goes deep into the matrix network. However, the reaction of the alcohol amine bonding agent with AP releases ammonia, resulting in holes and cracks in the solid propellant [6]. To avoid such problems, TEA·BF_3_ and composite bonding agents were developed. TEA·BF_3_ is a stable complex formed from Lewis base triethanolamine and Lewis acid boron trifluoride. Tang et al. [7,8] evaluated the coating effects of TEA·BF_3_, TEA, MAPO, and other bonding agents on AP and Al in a hydroxyl butyrate system and found that TEA·BF_3_ could be enriched on the surface of fine AP and Al particles and remained highly active. MAPO displayed an adverse synergistic effect, resulting in a significant increase in slurry viscosity. Shen et al. [9] complexed an alcohol amine bonding agent with a neutral bonding agent to retain the adsorption effect of the alcohol amine bonding agent on AP, while enhancing coating ability for nitramine. The complex increased the elongation at high temperature, normal temperature, and low temperature by 35–50%.

### 2.2. Polyamine Bonding Agent

Polyamine bonding agents were first used in PU propellants by Oberth [10,11,12]. Unlike alcohol amine bonding agents, they contain multiple amino groups or amine groups. The general structure of the polyamine bonding agent is shown in Table 1.

Tepan, the reaction product of tetraethylenepentamine and acrylonitrile, and Tepanol, the adduct of tetraethylenepentamine, acrylonitrile, and 2,3-epoxypropanol (glycidol), are typical polyamine bonding agents. Thomas et al. [14] was able to increase the ductility and ultimate strength of the TiO_2_/AP composite propellant by 181% and 7%, respectively, using Tepanol as the bonding agent. Pereira et al. [15] successfully applied Tepanol and castor oil as the filler–binder bonding agent (FBBA) by factorial design to improve the modulus and hardness of the propellant and discovered that the best mechanical properties and processability could be obtained with the NCO/OH molar ratio of 0.86, FBBA total concentration of 0.25, and CO:Tepanol mass ratio of 0.75.

### 2.3. Titanate Bonding Agent

Titanate bonding agents are a class of additives with Ti as the active center, which can be classified as monofunctional, chelating, and coordinating titanate bonding agents [16]. In particular, chelating bonding agents are usually used in nitramine propellant systems because it can construct the polar rings suitable for HMX and RDX particles and are resistant to hydrolysis. Li et al. [13] synthesized a series of long-chain alkyl chelating difunctional titanate bonding agents that could improve the slurry leveling properties and mechanical properties of the nitrate plasticized polyether propellant. A bonding agent, PA-15, containing oxalic acid group and short long-chain was developed, which could reduce the yield value of the propellant slurry by 36–55%, and increase the high tensile strength of the propellant to 0.9 MPa. Similarly, Lin et al. [17] prepared a series of dual-functional chelating titanate bonding agents that could improve the rheological properties and mechanical strength of the HTPB propellant. DLT-12 containing thee lactic acid group and n-dodecyl chain displayed the best comprehensive performance with the tensile strength of 1.83 MPa and a yield value as low as ~17 MPa.

### 2.4. Borate Ester Bonding Agent

Borate ester bonding agents (BEBA) take the boron atom as the structure center, and the molecular chains extending around are usually terminated with –OH. The boron atom is an electron deficient configuration surrounded by the B–O bond. It can actively react with the hydroxyl group of the binder via the transesterification reaction (Figure 2), and thus the bonding agent can penetrate into the binder network effectively. In addition, the *sp^2^* hybrid orbit of the boron atom forms stable N→B and O→B coordination bonds with the nitrogen and oxygen atoms in the NO_2_ electron donor group of nitramine (Figure 3), which enhances the coating of the bonding agent on the surfaces of inert nitramine particles. The terminal -OH reacts with the isocyanate group in the binder to form highly polar group –CONH that can form hydrogen bond with the –NO_2_ in nitramine particles due to the inductive effect. Therefore, an improved interface bonding ability takes shape between the oxidant filler and binder [18,19,20].

At present, the application of BEBA has been focused on the performance improvement of nitramine in the HTPB propellant. Several studies have proven that borate ester bonding agents can strengthen the anti-dewetting ability of nitramine particles and improve the high and low temperature mechanical properties of the propellant. Cui et al. [21] synthesized the alkanolamine intermediates respectively containing alkyl (–CH_3_), long carbon chain (–CH_2_CH_2_CH_2_CH_3_), cyano (–CH_2_CH_2_CN), and carbonyl groups (–COCH_3_) to construct the target borate ester structural bonding agents of the HTPB propellant. The uniaxial tensile test revealed that the bonding agents containing strong polar groups, –CH_2_CH_2_CN (BA-3) and –COCH_3_ (BA-4), greatly enhanced the mechanical properties of the propellant. The propellant with BA-3 displayed the smallest adhesion index (*ε_b_*/*ε_m_* = 1.07) and high tensile strength (*σ_m_* = 0.905 MPa) simultaneously.

With the demands on molecular weight and active sites of the bonding agent, the polymerization monomer was introduced into the structure as the interconnecting or external monomer (Figure 4).

Zhang’s group [22,23,24,25] synthesized different types of BEBA based interconnecting monomers and external monomers (Table 2). Interconnecting monomers are the polyether or polyester chains with low molecular weights that link two borate ester structures. External monomers are diethyl alcohol amine derivatives including different functional groups. By adjusting the functional groups in the external monomer and addition ratio of the bonding agent, they analyzed and discussed the interaction between the components and the performance of the HTPB propellant. The results suggested relatively prominent effects of polybutylene adipate and N-(2-cyanoethyl) diethanolamine on the properties of the HTPB propellant as the interconnection monomer and external monomer, respectively. The functional group (–CN) provided by the external monomer could enhance the molecular interaction with RDX due to the coordination bonds formed between B and N and O atoms. The ester group can enhance the affinity with the nitroamine oxidant, and the amino group introduced in the BEBA structure can form ammonium salt ionic bonds with AP. These interactions and reactions provide a strong interfacial interaction between the filler particles and the binder matrix, thus improving the mechanical properties of the propellant. Li et al. [26] introduced fluorine-containing functional groups into a diborate ester bonding agent linked by polyether segments to improve the surface activity and stability of the bonding agent (LBA–402–1,2,3). It was found that the F component in the bonding agent significantly reduced the viscosity of the slurry due to the optimal chemical activity, and the mechanical strength of the propellant was positively correlated with the content of –CN. The bonding agent, LBA–402–2, increased the ultimate strength and elongation of the grain by 11.4% and 32.9%, respectively, under the optimized condition and decreased the slurry viscosity by 20%, which improved the performance and process performance of the slurry and the particles.

However, borate ester bonding agents are prone to hydrolyze, thus its bonding performance is greatly affected by the environmental humidity. This shortcoming is inconducive to its storage and process. Therefore, Wang et al. [27] designed and synthesized a ring chelated borate ester bonding agent with a large volume to increase the steric hindrance and slow down the hydrolysis. The cyclic borate ester, especially the one with a five-membered ring, exhibited better water resistance than the chain borate ester. The substituents on the ring could stabilize the cyclic borate ester. The agent with a five-membered ring (0.2%) increased the tensile strength by 80% and extended the half-life time to 27 days in saturated water vapor.

### 2.5. Aziridine Bonding Agent

Aziridine bonding agents usually contain an aziridine ring and a polar group, C=O or P=O. For example, Hx-752, Hx-868, and TAZ contain C=O, and tris-(2-methylaziridinyl) phosphine oxide (MAPO) and its derivatives contain P=O (Table 3).

Aziridine bonding agents usually act on the surface of AP. They can be catalyzed by AP to undergo the ring-opening self-polymerization reaction, and thus is coated on AP via hydrogen bonding with multiple active sites, as shown in Figure 5. The hydrogen bonding between the aziridine bonding agent and AP requires the catalyzed ring opening reaction. There is no hydrogen bonding interaction with RDX and HMX, and thus showed no effects on the interaction between the nitramine oxidant and binder [29].

MAPO, TAZ, and HX-752 could all significantly enhance the mechanical properties of the HTPB propellant. The elongation of AP/Al/HTPB propellants can reach up to 23–36% at low temperature (−40 °C) and room temperature (20 °C) within a high strength range of 1.54–2.74 MPa [28,30]. Jiao et al. [31] simulated the adsorption of alcohol amines (TEA, etc.) and aziridines (MAPO, etc.) on Al/Al_2_O_3_ solid fillers in the HTPB propulsion system by molecular dynamics simulation using the COMPASS force field. They found that the performances of bonding agents were in the order of TAZ > MAPO•HAC > MAPO > HX-752. The binding energies between TAZ and Al crystal planes are as high as 115 kJ·mol^−1^ and the elastic modulus can be restored to about 55 GPa, similar to that of Al itself. The high interfacial adsorption energy and elastic modulus manifest the good mechanical properties of the propellant.

Hori et al. [32] prepared a bonding agent, MT_4_, by the reaction of two moles of MAPO with 0.7 moles of adipic acid and 0.3 moles of tartaric acid. The bonding agent formed a solid layer around AP. The FTIR analysis revealed a O=P–NH– structure in the bonding agent, in which the O and H could form hydrogen bonds with AP, respectively (Figure 6).

Ali et al. [33,34] prepared a long-chain bonding agent, MSA, with carboxylic acid and MAPO. They experimentally found that the mixture of MSA and a traditional MAPO bonding agent could increase the modulus and strain of the HTPB composite solid propellant and the mixture of 0.1% MAPO and 0.2% MSA exhibited the best performance with the Young’s moduli of 31.6 kg/cm^2^ and 81.66 kg/cm^2^ at room temperature (25 °C) and low temperature (−40 °C), respectively. The bonding agents greatly improved the yield performance of the propellant. Subsequently, to graft the hydroxyl group to MAPO, the group synthesized another new long-chain bonding agent, MST (poly (isopropyl amine N (2-methyl-1-aziridinyl) phosphine oxide octadecanoate)-co-(isopropyl amine tartarate)), and studied its effect on the mechanical properties and burning rate of the HTPB composite propellant. It was found that the MST bonding agent could significantly improve the mechanical properties and aging properties of the composite propellant with the maximum strain almost twice as high as that of the traditional MAPO formulation. The substitution of MAPO with MST reduced the burning rate from 7.02 mm/s to 6.44 mm/s, which created a more stable combustion process and weakened the influence of the pressure factor on burning rate.

### 2.6. Hydantoin/Triazine Bonding Agent

Hydantoin/triazine bonding agents are widely used in HTPB and nitramine propellants. They are nitrogen-containing five membered heterocycles (hydantoin) or six membered heterocycles (triazine isocyanurate adducts) containing –CONR. Hydantoin and triazine can also be used separately. The –NH– in the ring of hydantoin and triazine bond is strongly acidic, and the nitrogen atom with negative charge becomes an electron donor [35]. Therefore, the nucleophilic reaction can take place on the nitrogen atom to form corresponding hydantoin/triazine bonding agents. Consaga [36,37] introduced various substituents such as alkyl chain, alcohol hydroxyl group, carboxyl group, and mercaptan group into the cyclic molecular framework of hydantoin and s-triazine isocyanurate adducts, as shown in Figure 7, which opened an avenue to the synthesis of hydantoin/triazine bonding agents. Pires et al. [38] prepared 1,3-bis (hydroxymethyl) 5,5-dimethyl hydantoin that has been widely used in HTPB propellants. The large numbers of nitrogen heterocycles and active sites for the grafting of different polar groups of hydantoin/triazine complex bonding agent make it applicable to a variety of oxidants including CL-20 [39].

Hydantoin/triazine bonding agents have a cyclic structure similar to nitramine oxidants such as RDX. With the advantages of low internal stress and induction effect, the hydrogen bonds formed between the terminal functional group and polar oxygen atom can significantly improve the mechanical properties of propellant. Li et al. [40] coated RDX with a hydantoin/triazine bonding agent by the water/solution method. It was observed that the bonding agent formed an obvious coating layer on the surface of RDX, and the formation of hydrogen bonds was verified. The impact sensitivity analysis revealed that the characteristic drop heights were raised from 26.2 cm to 30.7 cm, and the bonding agent could improve the mechanical sensitivity of RDX. Brzic et al. [41,42] evaluated tris(2,3-epoxypropyl) isocyanurate (TEIC) and the mixture of 1,3,5-triglycidyl isocyanurate (TGIC) and N,N′-diglycidyl-5,5-dimethylhydantoin (DMH) with different contents as the bonding agent of the CTBN/AP composite propellant. They found that 0.5% TGIC in the mixture could increase tensile strength 97.7% and simultaneously decrease the strain at a maximum load value of 16.6%. TEIC of 0.1 wt.% exhibited the best performance with the increases in the strength and strain of 4.62% and 14.02%, respectively. Yao et al. [43,44] grafted a polyether molecular chain segment onto the amide ring of isocyanurate (Figure 8) by taking the advantages of the hydrogen bond formed between polyether polyol and solid fillers to prepare a novel polymer bonding agent. The XPS analysis indicated that more hydroxyl groups and cyano groups in the bonding agent were beneficial to the wettability of HMX, reflected with the high coating degree of 33.4%. In addition, the bonding agent showed no inhibition effect on the thermal decomposition behavior of HMX. The final ultimate tensile and elongation of the propellant were increased 19% and 213%, respectively, in the maximum extent. Pan et al. [45,46,47,48] prepared a dendrimer bonding agent (DBA) with terminal groups of –COOCH_3_, –CN, and –OH using polyamide (PAMAM) as the dendrimer skeleton. XPS and IR analyses showed that the bonding agent could well adsorb on the surface of the solid filler, and introduction of –CN improved the coating degree. The spectral peak shifts suggest that the N–H and N–O bonds in the solid fillers formed hydrogen bonds with –COO– or –CN in bonding agents.

### 2.7. Organosilane Bonding Agent

The molecular structure of silane bonding agent is YR-SiX_3_, where Y- is a non-hydrolyzable organic functional group, and X is a highly hydrolyzable group such as halogen, amine, alkoxy, and acyloxy. –Si–O– is prone to forming hydrogen bonds that can enhance the interfacial interactions of composite materials. Functional groups such as hydroxyl, amino, and cyano groups are usually introduced into the silane bonding agent [49], which can act on the adhesive matrix, and RDX and HMX, respectively.

In a propellant system, oxidant particles such as AP and AN tend to absorb moisture. The silane coupling agent can consume the absorbed water and enhance the interfacial binding due to the advantage of its water reactivity. Specifically, the siloxy group first hydrolyzes with the absorbed H_2_O to form –Si(OH)_3_, and the hydroxyl group is then dehydrated and condensed to form oligomers that are tightly attached via hydrogen bonds to form an adsorption modulus layer on the PA surface for the interfacial bonding (Figure 9) [50]. The silane coupling agent can usually improve the thermal stability and moisture stability of propellant [51,52,53].

Zhang et al. [54] coated the KH792 silane coupling agent onto the surface of phase-stabilized ammonium nitrate (PSAN) to form an enhanced layer via hydrogen bonding. The coated PSAN obtained by the coating pretreatment process had high adhesion, which increased the tensile strength from 0.40 MPa to 0.66 MPa and reduced the contact angle by half. Grau et al. [55] coated nitrocellulose with an organosilane connected by 3-(triethoxysilylpropyl) isocyanate. Isocyanate reacted with the –OH in nitrocellulose and the hydrolyzed triethoxysilanes formed cross-linked a siloxane silanol surface following the traditional mechanism. The whole reaction process showed no effect on the properties of nitrocellulose, but putting a layer of “waterproof raincoat” on nitrocellulose, which greatly improved the wettability of nitrocellulose. Zhu et al. [56] modified the nitroguanidine propellant with KH550, which improved the mechanical properties of the propellant with 34.22% and 11.5% increases in the impact strength and tensile strength, respectively, at low temperature (−40 °C). In addition, the KH550 coating caused a very small thermal decomposition peak shift, indicating that the bonding agent did not affect the thermal stability of the propellant significantly. In the latest research of Wang et al. [57], bonding agent BAG-ZD was prepared by the reaction of polyethylene glycol and 3-isocyanate propyl trimethoxysilane to improve the interfacial properties of the AlH_3_ propellant. After the post-processing, the surface of AlH_3_ still showed strong signals of Si and O with no particle aggregations observed, indicating that bonding agent did not affect the preparation processing. Meanwhile, the tensile strength and maximum elongation of the propellant were both increased by about 40%.

### 2.8. Neutral Polymer Bonding Agent

With the development direction of propellant to high-energy nitrate plasticized polyether (NEPE) propellants, two problems in the application of the alkaline small molecular bonding agent to the NEPE propellant have emerged: (1) the alkaline bonding agent can cause the degradation of the components with nitro or azide groups; and (2) the cohesive energy density of a polar plasticizer system is similar to that of nitramine particles, which can compete with the bonding agent for the adsorption sites on the surface of energetic particles. Bruenner et al. [58] found that the neutralization of Tepan could reduce the basic amino groups and thus improve the interfacial bonding. Kim et al. [59,60] then reported a neutral random copolymer containing 40 wt% acrylonitrile or acrylamide as the main chain and no less than three hydroxyl groups in the molecule, and named it the neutral polymeric bonding agent (NPBA) (Figure 10). The multiple action sites, –CN and –OH, on the neutral polymer can closely connect the nitramine particles and adhesive. They also demonstrated that 0.2% NPBA could produce a five times strength in the HMX/PEG-4500/NG/BTTN system. The NPBA also displayed excellent temperature dependent phase separation characteristics.

Because of the advantages of a neutral polymer bonding agent on the controllable molecular structure and molecular weight, it has become a hot topic in the research of bonding agents. Zhang et al. [61,62] prepared a high molecular weight (>10,000) block neutral polymer bonding agent by RAFT polymerization using acrylonitrile (AN), ethyl acrylate (EA), and hydroxyethyl acrylate (HEA) as the raw materials. A larger alkyl segment of the acrylate makes the solubility of the bonding system significantly higher than that of NPBA. The polymer bonding agents with higher molecular weights and higher glass transition temperatures can greatly improve the mechanical properties of the propellant at room temperature and high temperature. Lin [63,64] introduced NPBA into the TATB-based PBX explosive composite system to enhance the interfacial bonding between the TATB crystal and fluoropolymer. The XPS analysis confirmed the hydrogen bonding between TATB and NPBA, which improved the storage modulus, mechanical strength, and elongation at break of the explosive system. The research group also found that the mixture of NPBA and graphene could significantly improve the creep resistance of TATB-based PBXs. The compressive strength and tensile strength of PBXs filled with 0.5 wt% graphene and 0.1 wt% NPBA were increased by 13.1% and 49.2%, respectively, compared with those of the TATB based PBXs without the adhesives. Landsem et al. [65] prepared a regular NPBA and modified one, NPBA-OMe, with lower molecular weight by free radical polymerization. Both displayed low dispersion coefficients and doubled the tensile strength and elastic modulus of the HMX/GAP/energetic plasticizer (BuNENA) composite propellant. Zhou et al. [66] prepared alkynyl NPBA (BA-1) with higher isocyanate and alkynyl contents than those of a conventional NPBA (BA-0) molecular segment by a three-step method shown in Table 4. The alkynyl NPBA improved the internal frictional resistance of the GAP molecular segment movement and the interface bonding performance between GAP binder and CL-20. As the alkynyl content in the bonding agent increased, the maximum tensile strength and initial modulus of the GAP/CL-20 propellant reached 1.24 MPa and 2.79 MPa, respectively.

With the progress of NPBA research, it is necessary to discuss the sequence and phase separation characteristics of NPBA for the synthesis of efficient bonding agents with uniform structure under controllable conditions. Zhang et al. [67] tuned the feeding mode of the NPBA synthesis by the free radical copolymerization equation, investigated the thermal decomposition kinetics of the initiator and the relative molecular weight of NPBA, and optimized the amounts of initiator and chain transfer agent to prepare the NPBA bonding agent with uniform structure and high quality. They found that the relative molecular weight of NPBA decreased and the hydroxyl value slightly increased with the increase in the amount of chain transfer agent. Chen et al. [68] studied the influences of reactivity ratio and monomer dosage of NPBA monomers on the sequence composition of bonding agent by Monte Carlo simulation. The results were in good agreement with the experimental results of mechanical properties of the nitrate plasticized composite solid propellant formulated with NPBA. Because of its characteristics of high temperature dissolution and low temperature phase separation, the aggregation behavior of the polymer molecular chain in solution is between macroscopic and microscopic levels, which needs to be described in the mesoscopic scale. Yu et al. [69] calculated the order degree of the bonding agent system by the mesoscopic dynamics simulation method. The results verified that the phase separation degree of NPBA in the energetic plasticizer/prepolymer system increased at low temperature, which referred to the molecular bridge that formed at the interface. Wang et al. [70] found that NPBAs with high contents of AN were prone to cooling phase separation in inert solutions, consistent with the changing trend of their interactions with AP under different conditions. Therefore, they established a molecular dynamics model of cooling phase separation behavior of NPBA in a NEPE propellant to predict the effects of plasticizer ratio, the concentration, and molecular weight of NPBA, as well as the terpolymer composition of NPBA on the phase separation of NPBA in the mixed slurry, optimizing NPBA, and tuning the properties of the slurry.

### 2.9. Other Types of Novel Bonding Agents

In recent years, the research on bonding agents has followed the demand for propellants. Many new bonding agents such as alkyl amine bonding agents, nitrile-butadiene rubber (NBR) bonding agents, and hyperbranched polyether bonding agents have been innovated and various bonding technologies have been developed to directly graft the bonding groups onto the binder surface. We have [71,72] reported a variety of amine bonding agents containing long alkyl chains. The excellent binding performances of the new bonding agents were confirmed by RDX, molecular dynamics simulation, and XPS. In addition, we found that a stronger interfacial interaction could be gained with more polar functional groups in the long chain. Xu et al. [73] studied the effects of nitrile butadiene rubber bonding agents on the properties of the CL-20/GAP propellant. The SEM and DMA analyses revealed that CL-20 was well coated, the elongation at break of propellant film was increased by 10%, and the maximum elongations at high temperature and low temperature were increased by 20–30%. Deng et al. [74].and Zhao et al. [75] respectively compared the binding effects of different hyperbranched polyether bonding agents (BA3, CBPEs, and BPEs) and small molecule bonding agents on solid fillers. The adhesions between BA3/HMX and CBPE-10 and 10/CL-20 reached 67.27 mN⋅m^−1^ and 105.37 mN⋅m^−1^, stronger than those between the small molecule bonding agent/solid filler system.

Directly grafting bonding groups on the binder matrix is also a feasible solution to avoid mutual solubility of the polar bonding agent and plasticizer. Sun et al. [76] synthesized a glycidyl azide polyol energetic binder grafted with a –CN bonding group and simulated its performance by molecular dynamics simulation. They demonstrated that the strong inductive effect between nitro and cyano groups increased the binding energy of the binder/RDX model propellant to 821.58 KJ⋅mol^−1^.

With the attractions of various biomaterials and the enlightenment of bionics, Lin and He et al. [77,78,79,80,81] proposed modifying the interfacial properties of energetic crystals such as TATB by coating them with polydopamine (PDA) as a bonding agent. The interfacial layer with a graphite-like structure increased the tensile strength up to 15%. It was then conjectured that there was a physical ‘interlocking block’ model on the surface favorable for the interfacial bonding. Different linear polymers including GAP, PEG, and PTMEG, and hyperbranched polymers (HBPs) such as hyperbranched polyurethane were also grafted onto the surface of the PDA coated shell based on the principle of marine mussel, which enormously increased the maximum tensile strength and compressive strength of TATB of 70% and 55%, respectively.

## 3. Progress on Evaluation Method of the Interfacial Interactions of Bonding Agents

“Interfacial interaction ability” is the evaluation index for the application performance of a bonding agent, which refers to the ability of a bonding agent to form a tight junction layer on the filler surface via different interactions such as hydrogen bonding and van der Waals forces. The ability affects the overall performance parameters of the composite solid propellant. In general, the stronger the interfacial interaction ability, the better the performance of the propellant from processing to application comprehensively. The methods to qualitatively and quantitatively evaluate the interfacial interaction ability of a bonding agent will be discussed in the next section.

### 3.1. Basic Qualitative Methods

#### 3.1.1. Mechanical Analysis

The simplest qualitative evaluation method in most studies is to assess the interfacial interaction ability with the mechanical properties of the propellant sample by the quasi-static uniaxial tensile test and DMA test. The tensile test obtains the tensile strength *σ_m_*, elongation (fracture elongation *ε_b_*, maximum elongation *ε_m_*) and adhesion index (*ε_b_/ε_m_*). The bonding agents that have high *σ_m_*, *ε_m_*, and *ε_b_* are generally considered optimal. The DMA test measures the loss factor tan δ, the damping of the material. The decrease in loss factor tan δ indicates that the interaction between the filler and the matrix is improved.

Xie et al. [82], Zhang et al. [83], and Xu et al. [73] studied the effects of bonding agent content on the mechanical properties of BAMO-THF solid propellants, PBT based reduced smoke propellants, and CL-20/GAP/TDI propellants, respectively, by the uniaxial tensile test method. They measured the *σ_m_*, *ε_m_*, and *ε_b_* of the specimens with different bonding agent contents at high temperature, normal temperature, and low temperature and found that the bonding agents containing amino, borate ester, and hydroxyl groups could enhance the tensile strengths and elongations of the BAMO-THF solid propellants and PBT based reduced smoke propellants more evenly. Nitrile butadiene rubber bonding agents such as BAG-1, 2, and 3 could increase the *ε_m_* and *ε_b_* of CL-20/GAP/TDI propellants more than 10%. Xu et al. [73] also measured the loss factor tan δ of the propellant with BAG-1, 2, or 3 and discovered that the propellant showed a decreasing trend in the temperature range −40–10 °C. Landsem et al. [84] studied the effects of NPBA on the interfacial interaction in the presence and absence of an isocyanate curing agent by the DMA test. The results revealed that NPBA could integrally reduce the loss factor curve of dual cured HMX-GAP-BuNENA in the temperature of range −60–20 °C.

#### 3.1.2. Conventional Structural Characterization Methods

Fourier transform infrared spectroscopy (FTIR) and nuclear magnetic resonance (NMR) are commonly used structural characterization methods to explore the interfacial interaction mechanism. The reaction or bonding interaction at the interface can be identified with the changes in the characteristic peak intensity, chemical shift, and binding energy of the bonding agent groups.

Petković et al. [85] synthesized a series of 1,3,5-trisubstituted isocyanurates and characterized them as bonding agents by FTIR and ^1^H NMR. The disappearance of characteristic bands and the peak shifts of O–H, N–H, and C=O of the bonding agent BA_1_ and AP in AP/CTPB/HTPB confirmed strong hydrogen bonds between AP and bonding agent BA_1_, and the decarboxylation reaction of CTPB with the bonding agent. Azoug et al. [86] considered a propellant as three segments with different relaxation times. By measuring the changes of fluidity of each segment under load deformation with the spin relaxation time T_2_ of the propellant by ^1^H NMR, they found the bonding of FBBA significantly decreased the fluidity of the segment near the packing.

### 3.2. Semi-Quantitative Surface Analysis Methods

#### 3.2.1. X-ray Photoelectron Spectroscopy (XPS)

XPS can provide the mass fraction of N on the surface of a bonding agent, which can then be used to calculate the coating degree of the oxidant particle by the bonding agent using the following equation.
*R =* (*N*_0_ − *N_x_*)/*N*_0_(1)
where *R* is the degree of coating; *N*_0_ is the mass percent of N atom on the surface of the uncoated sample; and *N_X_* is the mass percent of the N atom on the surface of the coated sample.

Zhang et al. [87] coated RDX with triethanolamine and the neutral polymer materials, LBA-201 and LBA-603, and measured the coating efficiencies by XPS. LBA-201 with high hydroxyl and nitrogen contents obtained the maximum coverage of 70%, which could significantly improve interfacial properties. Liu et al. [71] calculated the coating degrees of three bonding agents, ADODD, DDID, and DHAP on RDX with the percentage N1s XPS peak intensity and the peak separation of N1s binding energy and obtained consistent results. DHAP showed good interaction abilities with RDX with the coating degrees of 58.65% and 18.06%, respectively, consistent with the results of the mechanical experiment. Yang et al. [88] measured the coating degree of the polyene polyamine bonding agent (LBA-1) on RDX by XPS to be 70%. The coated RDX improved the mechanical properties of the CMDB propellant with 8% increase in the elongation due to the excellent interaction between the bonding agent and RDX.

#### 3.2.2. Micromorphological Analysis

Scanning electron microscopy (SEM), a traditional surface characterization method, can only provide the morphological image of bonding agent coated on the filler particles or the fracture section morphology of propellant for the analysis of interfacial interaction. The imaging is greatly affected by the quality of the specimen. To visualize the interfacial bonding effect of bonding agent in the propellant more accurately and clearly, in-situ loading scanning electron microscopy (in-situ SEM) was introduced.

In situ SEM that combines SEM imaging technology with in situ tensile loading can visualize the evolutions of morphology and the internal structure of a solid propellant under loadings for a period of time and thus obtain images of the failure effect of interfacial bonding. The influences of bonding agent on the interfacial bonding of the propellant can then be evaluated with the SEM images and tensile curves. Therefore, in situ SEM is considered as an improved surface analysis method. Zhao et al. [89] proposed a uniaxial tensile in situ SEM method to analyze the effects of MAPO content on the mechanical properties of the HTPB propellant. The design was able to image the initial compact morphology of the specimen with MAPO due to the coating effect of the bonding agent. In addition, MAPO reduced the damages and cracking degree of the specimen mostly around the large AP particles. The stress–strain curve revealed that the modulus of the matrix was increased by 172%. Benedetto et al. [90] and Van et al. [91] studied the effects of temperature and tensile strain rate on the failure mechanism of the HTPB/AP/Al system with Tepanol as the bonding agent. The temperature was varied to −54 °C, 25 °C, and 40 °C at the tensile rate of 150 μm/min and different tensile rates of 30 μm/min, 150μm/min, and 750 μm/min were tested at 25 °C. The SEM images illuminated the nucleation and growth processes of cracks and holes during the failure process of the propellant sample, and that the low temperature significantly affected the stiffness of the propellant specimen. Despite the presence of Tepanol, the propellant broke directly, showing a different failure behavior from the chewing gum tension at room temperature and high temperature. It was also observed that higher tensile rates resulted in short drawing phases.

Atomic force microscopy (AFM) is another widely used surface characterization technique. It scans the surface of a material with a very small probe (<10 nm) and records the three-dimensional morphology of the material surface based on the change in the interaction force. Toulemonde et al. [92] observed the modulus in micro-scale by AFM and obtained the clearly edges of white filler particles, dark gray matrix, and light gray interphase. The Young’s modulus evolution curve showed that the interphase modulus was about five times of that of the matrix (5 MPa).

However, these studies only qualitatively characterized the failure surface of the solid propellant and failed to describe the change law of interfacial interaction in the propellant. In recent years, quantitative analysis of the interfacial interaction of the propellant by in situ SEM combined with digital image processing has become a research hotspot to explore the failure law of propellants.

The digital image processing converts the image obtained by in situ SEM into specific data by digital operation. The data can be used to describe the state change of the sample. At present, there are mainly two digital image processing methods for the study of propellant failure.

One method is to calculate the fractal dimension. Briefly, the SEM images are converted into black and white binary images and the fractal dimension value of each SEM image is calculated. The chaos of fractal dimension reflects the roughness of the image. Shi et al. [93] analyzed the tensile failure of NEPE propellant by in situ SEM. Based on the calculated fractal dimensions and other parameters, they proposed that the failure crack propagation law of propellant conforms to the Boltzmann function. The regression calculation suggested that the parameters of the sample with 0.25% NPBA were higher than those of the sample with 0.1% NPBA, and thus the interfacial interactions of the samples with high NPBA contents were stronger. Chen et al. [94] studied the effects of #22 bonding agent hydantoin content on the NEPE propellant by in situ SEM and digital image processing and obtained the evolution trends of fractal dimension at different bonding agent contents. With the increase in #22 bonding agent hydantoin content from 0.15% to 0.3%, the trend of fractal dimension (e.g., degree of disorder) became slower, indicating that the bonding agent improved the interfacial interaction and prolongated the smooth state of the propellant at the initial failure stage.

The second method is to collect the cloud images of strain and displacement. The deformation field is obtained by calculating the digital speckle image to describe the strain and displacement of the sample surface. Wang et al. [95] characterized the crack propagation and explored the failure mechanism of the HTPB propellant by in situ SEM and obtained the meso deformation field of the propellant by digital image processing. The interfacial interaction among the propellant components was quantitatively evaluated with the strain extremum near the crack tip. The results suggested that large extremum was accompanied by strong interactions. Wu et al. [96] investigated the failure of thee HTPB propellant linear interface by in situ SEM and obtained the cloud charts of strain extremum in the X and Y directions. They found that the local extremum in the propellant region increased sharply as the external strain exceeded 25%. Therefore, the particle dewetting occurred rapidly and finally affected the interface failure.

#### 3.2.3. Contact Angle

Contact angle, surface tension, and adhesion work are important parameters to describe interfacial interaction. These parameters reflect the wettability of a material and thus can be used to evaluate the strength of interfacial interaction. The contact angles of the filler particle and film spline of the propellant are usually measured by the modified Washburn method and Wilhelmy plate method, and the corresponding surface tension and adhesion work are then derived from the measured contact angles using Young’s equation. The relation of three indexes can be deduced based on Young’s equation as follows [97].
*Wa* = *γ*(1 + cos*θ*)(2)
where *θ* is the solid–liquid contact angle; *γ* is the liquid surface tension; and *Wa* is the adhesion work.

The Owens–Wendt–Rabel–Kaelble method (OWRK) developed based on the above relation is applicable to the calculation of the contact angle of each component in a composite solid propellant system. Zhou et al. [98] measured the contact angle of each major component of the nitramine-based propellant by both the Wilhelmy and Washburn methods and calculated the surface tension and adhesion work. They found that both of the bonding agents, PAN-b-PHEA and PAN-b-P(HEA-g-AEFC), exhibited excellent interactions on CL-20, HTPB, and some other components of the propellant. The ferrocene group in AEFC slightly decreased the surface tension of PAN-b-P(HEA-g-AEFC) with the value of Δγ = 5.28 mN/m, suggesting a weaker interaction. Zhao et al. [99] measured the contact angle of AP particles by the Washburn method and calculated the adhesion work between the AP and MAPO/HTPB matrix by OWRK. With the key parameters of critical debonding displacement calculated from the adhesion work, they successfully constructed a finite element model for the meso-debonding of the composite propellant interface. Qi et al. [100] simulated the binding effects of NPBA containing different functional groups on the HMX crystal surface and calculated the surface tension and adhesion work between NPBA and HMX by the contact angle method. The results showed that increasing the numbers of –OH and –CN groups on the chain segment of the polymer bonding agent could enlarge the adhesion work between molecular interfaces and cyano groups reduced the surface tension and improved the wettability, which were consistent with the experimental results. Sun et al. [76] analyzed the interfacial reinforcement of RDX after grafting –CN onto the GAP/HMDI adhesive by contact angle measurement and the creep resistance test. It was verified that there was an inducing effect between –CN and –NO_2_ in RDX that could enhance the surface binding effect of RDX.

### 3.3. Simulation and Calculation

In recent years, with the development of computational materials science, multi-scale and multi-scale simulation methods have been developed in both experimental and application stages. Computational materials science is of great significance for the study of energetic materials. It lowers experimental risks and costs, shortens the development cycle, and provides theoretical guidance for research.

Molecular dynamics (MD) simulation has been extensively applied to the evaluation of the interfacial interaction between the bonding agent and solid filler to reveal the relationship between the structure and performance of a composite propellant at the molecular level. In general, the adsorption behavior of the bonding agent can be discussed from the following aspects to analyze its interfacial enhancement mechanism.

First, by establishing the adsorption equilibrium structure and analyzing the concentration distribution of the oxidant and bonding agent in different directions to determine the migration direction and degree of the molecules, the progressive state of the bonding agent coated oxidant particles can be obtained and thus the degree of interfacial interaction ability can be calculated. Zhu et al. [101] simulated the equilibrium structure of a multicomponent system of (PEG/NG/BTTN)/BA/HMX/AP/PEG/N-100/HTPB/TDI by MD simulation. The concentration distribution curves of OA(X), OB(Y), and OC (Z) obtained by atomic motion trajectory revealed that the bonding agent BA and oxidant HMX tended to migrate to the interface. Zhang et al. [102] established an adsorption model of NPBA/PEG on a RDX surface, and simulated the compatibility of NPBA, PEG, and RDX, respectively. They discovered that the compatibility between NPBA and RDX was the best. NPBA more likely migrated to the surface of RDX than the plasticizer PEG and formed a binding shell to prevent the polar plasticizer from dissolving oxidant particles and damaging the interface bonding.

Second, the interfacial bonding can be evaluated by calculating the binding energy (*E*_bind_) between the bonding agent and the filler particles and the radial distribution function *g*(*r*) of the atom pairs in the system. Binding energy measures the magnitude of the interaction between two components. A large binding energy indicates a strong interaction. *g*(*r*) reflects the distribution of particles in the system. A large *g*(*r*) at a small r means a greater probability for two particles to become close with each other and thus a high probability of strong interaction.

*E*_bind_ and *g*(*r*) can be calculated using the equations below [19,103,104]:*E*_bind_ = (*E*_Oxidant_ + *E*_bonding agent_) − *E*_total_(3)
where *E*_bind_ is the binding energy between the bonding agent and oxidant; *E*_Oxidant_ is the energy of oxidant crystal face; *E*_bonding agent_ is the energy of bonding agent; and *E*_total_ is the energy of bonding agent/oxidant system.
*g(r)* = *N*_AB_/(*ρ4πr*^2^),(4)
where *N_AB_* is the number of particles B in the distance range from *r* to *r* + d*r* from the central particle A; and *ρ* is the average density of particles B.

Cui et al. [19] established surface interaction models of RDX with five BEBA derivatives, BEBA-1, 2, 3, 4, and 5 with different functions groups, respectively. The binding energy of BEBA-5 was the highest with the value of 1052.02 kJ⋅mol^−1^, suggesting that there was an inductive effect between its carbonyl group and the –NO_2_ of RDX at relatively low contents of oxygen atoms on the surface of RDX. By calculating the binding energy and radial distribution function *g*(*r*) at the NPBA and HMX crystal interface, Qi et al. [98] determined the relationship between the binding strength and the number of functional groups including –CN and –COOCH_2_CH_2_OH as well as steric hindrance, and concluded that the stronger polarity of the side group caused the stronger binding ability. Their work can effectively guide the molecular design of NPBA bonding agents. Zhang et al. [102] investigated the bonding mechanism of NPBA in the NEPE system containing PEG and nitrate plasticizer by MD simulation and discovered strong hydrogen bonding and van der Waals interaction between NPBA and RDX with the calculated binding energy and radial distribution function. They also demonstrated that binding energy was positively correlated with the elastic modulus by simulating the elastic coefficient matrix. Zhang et al. [103] calculated the binding energies and *g*(*r*) values between bonding agent (GAP_PDMH) and solid components (RDX, HMX, AP) of the GAP propellant by MD simulation. The results suggested that the *g*(*r*) value of the functional group was large enough to cause hydrogen bonding and strong van der Waals forces between the GAP_PDMH and solid components.

Third, the interfacial enhancement mechanism can be investigated by deriving bulk modulus, shear modulus, and Poisson’s ratio from the uniaxial tensile deformation and shear deformation of the system by static analysis of MS. These parameters can evaluate the stiffness and elastic deformation resistance of a material that reflect the strength of the system and the influence of bonding agent on the interfacial effect. It is expressed by Hooke’s law:(5)$$\left[\begin{array}{c} \sigma 1\\ \sigma 2\\ \sigma 3\\ \sigma 4\\ \sigma 5\\ \sigma 6 \end{array}\right]=\left[\begin{array}{cccccc} C11 & C12 & C13 & C14 & C15 & C16\\ C21 & C22 & C23 & C24 & C25 & C26\\ C31 & C32 & C33 & C34 & C35 & C36\\ C41 & C42 & C43 & C44 & C45 & C46\\ C51 & C52 & C53 & C54 & C55 & C56\\ C61 & C62 & C63 & C64 & C65 & C66 \end{array}\right]\left[\begin{array}{c} \varepsilon 1\\ \varepsilon 2\\ \varepsilon 3\\ \varepsilon 4\\ \varepsilon 5\\ \varepsilon 6 \end{array}\right],$$

Jiao [31] and Zhang et al. [102] respectively measured the elastic coefficient matrices of the different crystal planes of the Al/HTPB and NPBA/PEG/RDX adsorption systems by this method and compared the ductility of the adsorption systems with different crystal faces. It was found that the bonding agent increased the elastic modulus of the adsorption system and strengthened the adsorption interface.

Mesoscopic dynamics (MesoDyn) extends the research scope of the spatial level of MD. It describes and analyzes the evolution of mesoscopic space. Based on the mean field density functional theory, MesoDyn transforms a polymer into a coarse-grained Gaussian chain model and describes the movement of the Gaussian chain with the Langevin equation, which is of great significance to the study of complex fluid and polymer blends. Yu et al. [69] evaluated the degrees of phase separation of NPBA in BAMO, PEG, and NMMO prepolymer systems by simulating phase separation behaviors of NPBA bonding agents in energetic plasticizer/binder prepolymer by MesoDyn. The calculation results of ordering degree showed that NPBA is consistent with the design principle of compatibility at high temperature and phase separation at low temperature. Increasing the molecular weight of NPBA and decreasing the premixing temperature are beneficial to the phase separation behavior of bonding agents. Wang et al. [70] constructed a coarse-grained mesoscopic model to simulate the phase separation behaviors of NPBA in a nitrate ester plasticizer. The order degree analysis showed that high AN content was beneficial to the phase separation of NPBA, consistent with the influence law of AN content obtained by the mechanical experiments. The model can thus effectively predict the change in interfacial interaction ability and provide guidance for the optimization of various factors in a propellant system.

## 4. Conclusions

To improve the interfacial properties of the energetic composites, bonding agents of macromolecules, multi-functional groups, and multi active sites have gradually been developed for complex propellant systems. In addition to preventing the interface “dewetting” of the propellant, which causes poor mechanical properties, the bonding agent also significantly affects the aging performance, safety performance, storage performance, and explosive performance of the propellant system and thus has attracted more and more attention. Frontier interdisciplinary science, with the aid of the adhesion mechanism of bionic science, guides the composition and structure design of the bonding agent and forms the development trend of the interdisciplinary, which is conducive to the integration and innovation of bond agent design, synthesis, and application. To understand the effects of the bonding agent in depth, evaluation methods from micro- and meso-aspects such as binding energy, interfacial tension, and crack formation process have been further developed by intuitive data model and performance parameters. In the future, the research and development of a bonding agent should be inseparable from the improvement in performance evaluation methods. It is necessary to develop advanced detection methods and instruments to deeply understand the interfacial mechanism of the bonding agent and meet the development needs of propellants.

## Figures and Tables

**Figure 1 molecules-27-00340-f001:**
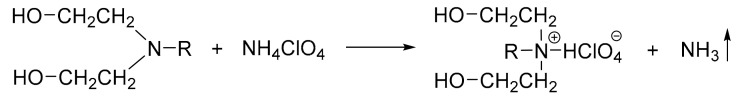
Bonding mechanism of the alcohol amine bonding agent.

**Figure 2 molecules-27-00340-f002:**
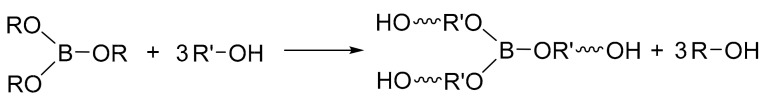
Transesterification reaction mechanism of BEBA.

**Figure 3 molecules-27-00340-f003:**
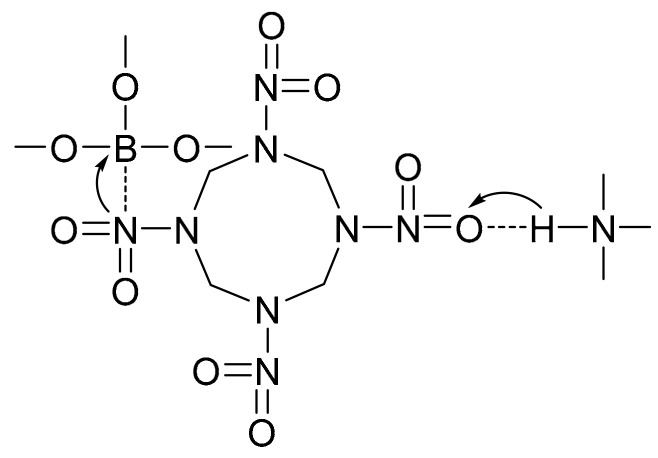
Bonding mechanism between BEBA and HMX.

**Figure 4 molecules-27-00340-f004:**
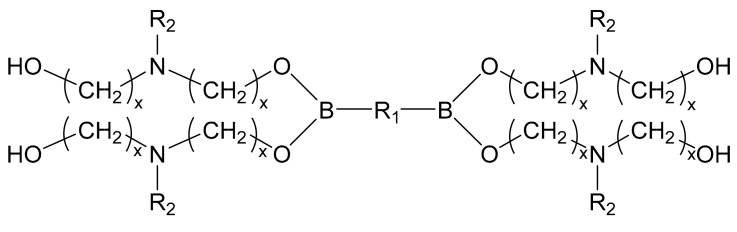
General molecular structures of dibasic borate ester bonding agents (R_1_-interconnecting monomer, and R_2_-functional group of external monomer).

**Figure 5 molecules-27-00340-f005:**
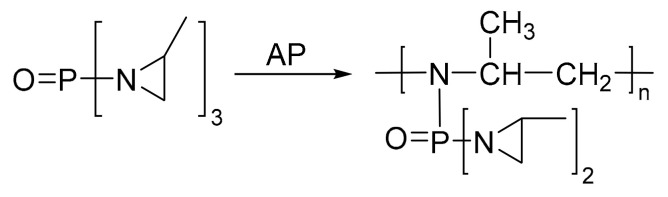
Mechanism of ring opening homopolymerization of MAPO.

**Figure 6 molecules-27-00340-f006:**
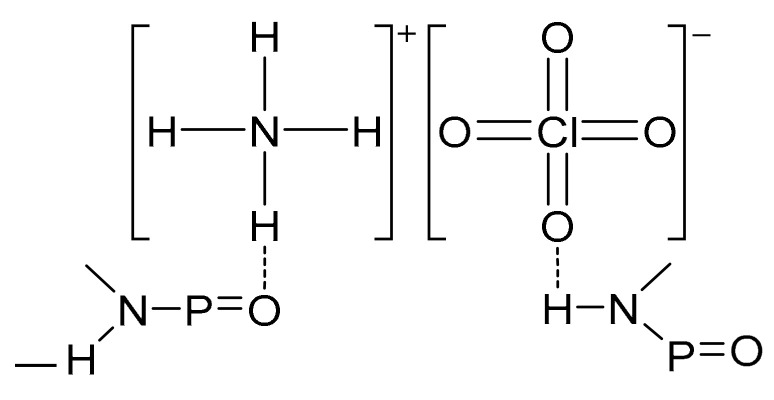
Bonding mechanism of MT_4_ and AP.

**Figure 7 molecules-27-00340-f007:**
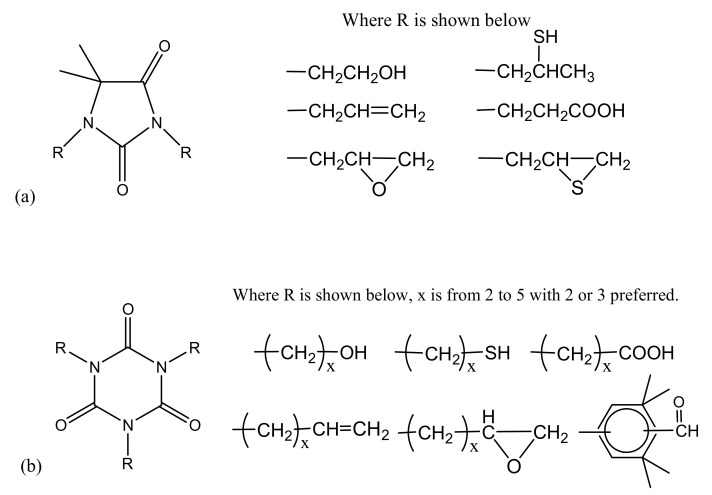
General chemical structures of hydantoin (**a**) and triazine (isocyanurate) (**b**) bonding agents.

**Figure 8 molecules-27-00340-f008:**
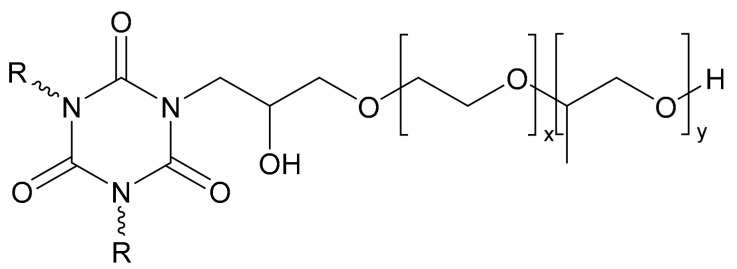
Polyether heterocycle amide bonding agent.

**Figure 9 molecules-27-00340-f009:**
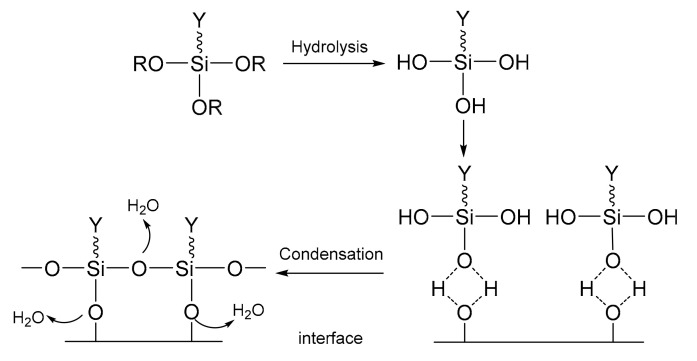
Action mechanism of the silane coupling agent.

**Figure 10 molecules-27-00340-f010:**
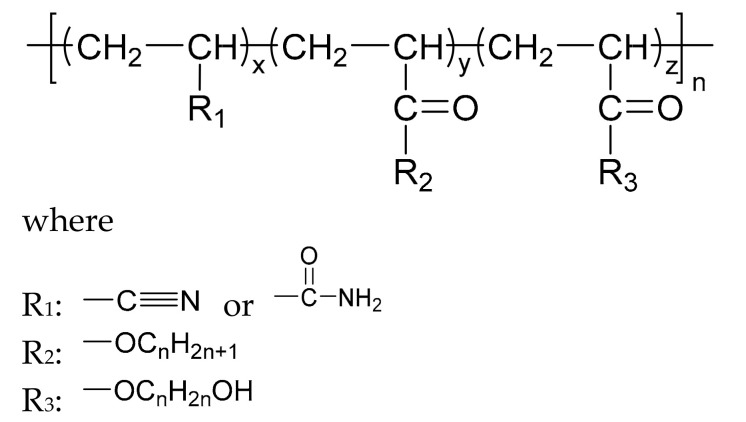
General structure formula of NPBA.

**Table 1 molecules-27-00340-t001:** General structures of common small molecule bonding agents.

Types	General Structure	Illustration	Application Cases
Alcohol Amine	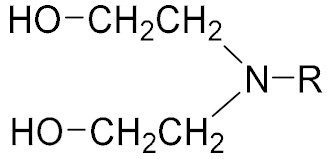	R: hydrogen, benzene ring, alkyl, ketone	[4]
Polyamine	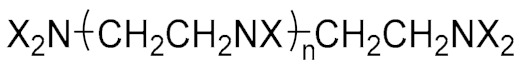	n:1–12, X: hydrogen, cyanoethyl, hydroxypropyl	[10,11,12]
Titanate	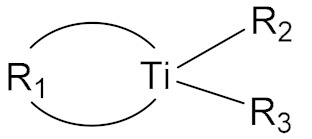	R_1_: bifunctional acid group; R_2_, R_3_: alkoxy or oleic acid group	[13]

**Table 2 molecules-27-00340-t002:** Interconnection monomers and external monomers of BEBA.

Interconnecting Monomer	Structure [22,25]	External Monomer	Structure [23,24]
N-Methyl-N,N-diethanolamine	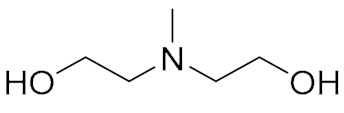	Polyepichlorohydrin	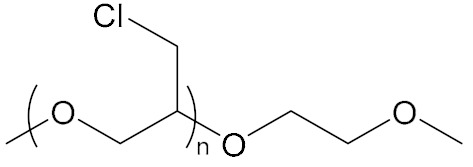
N-Butyl-N,N-diethanolamine	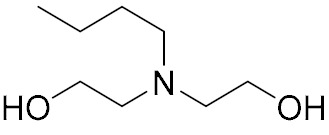	Poly(Propylene Oxide)	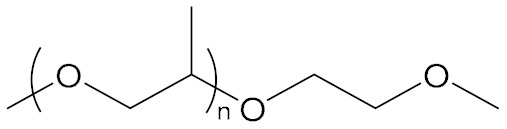
N-(2-Cyanoethyl)diethanolamine	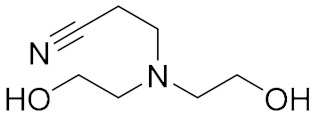	Poly(Ethylene Glycol Adipate)	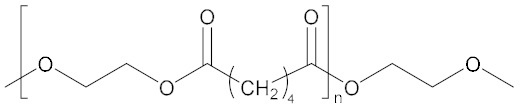
N,N-Dihydroxyl-3-aminmethyl Propionate	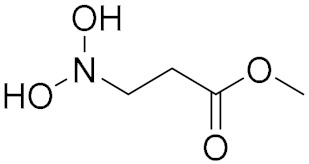	Poly(Butylene Adipate)	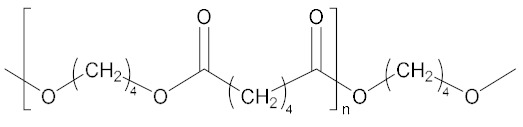
N,N-Dihydroxyethyl-3-amino methyl Propionate	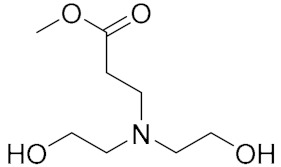	Polyethylene glycol	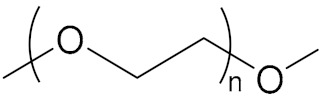

**Table 3 molecules-27-00340-t003:** Common aziridine bonding agents [28].

Name	Structure	Polar Group	Applicable Propellant System
Hx-752	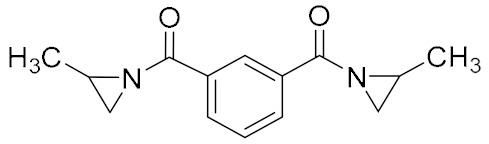	C=O	HTPB, PU
Hx-868	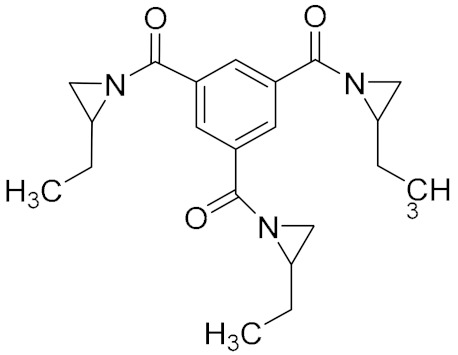	C=O	HTPB, PU
TAZ	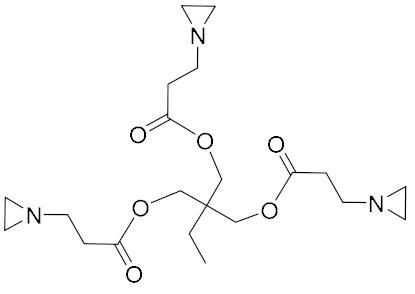	C=O	HTPB, PU
MAPO	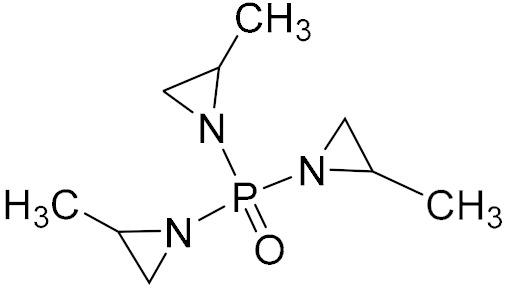	P=O	HTPB, PU

**Table 4 molecules-27-00340-t004:** Structures of the bonding agents mentioned in the literature.

Bonding Agent	Structure	References
NPBA	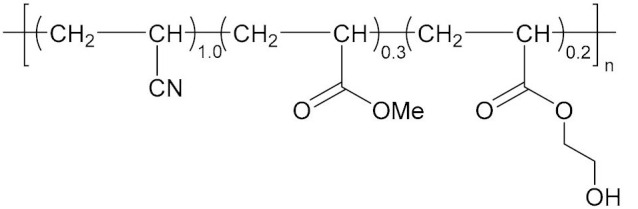	[65]
NPBA-OMe	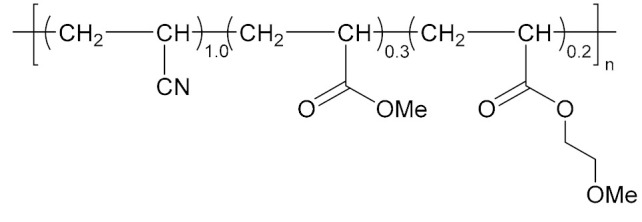	[65]
BA-0	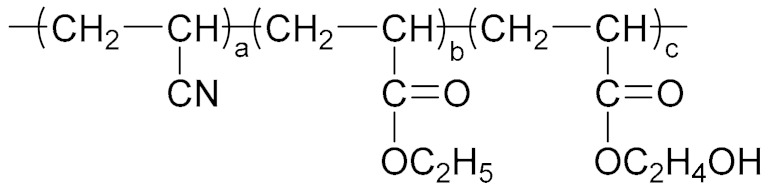	[66]
BA-1	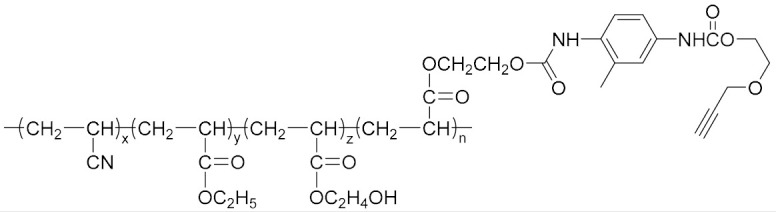	[66]

## Data Availability

Not applicable.

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
