# Peer review of "Research Progress of Bonding Agents and Their Performance Evaluation Methods"

_molecules, 2022, doi:10.3390/molecules27020340_

Round 1

Reviewer 1 Report

Dear Authors,

In my opinion, the reviewed article is an extensive and complete study on known modifiers of interfacial interactions used in fuels and PBX compositions. I find the attempt to explain the mechanisms of interactions and the presentation of qualitative and quantitative methods of assessing the effectiveness of these additives in improving the mechanical and explosive properties of energetic materials particularly valuable and useful for other researchers.

Regards

Author Response

Many thanks for the affirmation of the reviewer. We hope the paper has reference and guiding value for researchers in this field.

Reviewer 2 Report

This manuscript summarizes the use of the bonding agents and their performance evaluation methods. The review is generally well written, well organized and I think it is interesting for synthesis and materials science researchers working in the field of propellants, but it presents some critical issues that should be correct before its acceptance for publication.
The scope of this review is very specific and will be very interesting to the community of energetic materials. The authors write this manuscript as a summary of very few recent research papers and many of them were not cited in the journal format. References 2, 6, 8, 21, 29, 40, 41, 49-51, 61, 83, 87, 89, and 97 have no DOI numbers. Reference 7 was incorrect. For a better understanding of this manuscript, all the abbreviations should define their acronyms. In fact, this topic is very hot and interdisciplinary with synthesis and material science and many other research groups are working in recent years. As a result, this review asks them to cite more latest articles.

Author Response

Many thanks for the comments of the reviewer. What you mentioned is indeed the problems in our manuscript. We have made detailed revisions to the manuscript according to the problems. All of the DOI numbers have been supplied in the manuscript. Reference 7 has been checked and updated. All the abbreviations have been checked. They were defined as their acronyms or the abbreviation recognized by academia. By the way, we have supplied latest literature as much as possible. But you can see that some classic literatures are difficult to be replaced.
